# Accelerating UN Sustainable Development Goals with AI-Driven Technologies: A Systematic Literature Review of Women’s Healthcare

**DOI:** 10.3390/healthcare11030401

**Published:** 2023-01-31

**Authors:** Pin Lean Lau, Monomita Nandy, Sushmita Chakraborty

**Affiliations:** 1Brunel Law School, Brunel University London, Uxbridge UB8 3PH, UK; 2Brunel Business School, Brunel University London, Uxbridge UB8 3PH, UK; 3Independent Researcher, Manchester M13 0HX, UK

**Keywords:** women’s healthcare, artificial intelligence, UN sustainable development goals, SDG3, SDG5, gender equality, health equality, health sustainability

## Abstract

In this paper, we critically examine if the contributions of artificial intelligence (AI) in healthcare adequately represent the realm of women’s healthcare. This would be relevant for achieving and accelerating the gender equality and health sustainability goals (SDGs) defined by the United Nations. Following a systematic literature review (SLR), we examine if AI applications in health and biomedicine adequately represent women’s health in the larger scheme of healthcare provision. Our findings are divided into clusters based on thematic markers for women’s health that are commensurate with the hypotheses that AI-driven technologies in women’s health still remain underrepresented, but that emphasis on its future deployment can increase efficiency in informed health choices and be particularly accessible to women in small or underrepresented communities. Contemporaneously, these findings can assist and influence the shape of governmental policies, accessibility, and the regulatory environment in achieving the SDGs. On a larger scale, in the near future, we will extend the extant literature on applications of AI-driven technologies in health SDGs and set the agenda for future research.

## 1. Introduction

In 2015, the United Nations (UN) Member States adopted the UN’s 2030 Agenda for Sustainable Development. In a historic move, an international community has unequivocally committed to a global agenda for peace, people, and our planet through [1] the 17 Sustainable Development Goals (SDGs), which are an urgent call for action by all countries in a global partnership. They recognize that ending poverty and other deprivations must go hand-in-hand with strategies that improve health and education, reduce inequality, and spur economic growth.

In recent discussions among academics and policy makers, we surmise that much more attention has been devoted to the speed of achievement of the SDGs [2]. However, there are no detailed discussions about the inclusion of women’s health under the health SDG. To address this gap in the literature, we conducted an in-depth study through a systematic literature review (SLR). In this SLR we included not only maternal, sexual, and reproductive health and healthcare but also all incidental junctures such as access, cost, quality, treatment protocols, and the proliferation of sophisticated new and emerging technologies in biomedical interventions to examine our research question. 

The relevant UN SDGs in this study are represented by both SDG3 (generally encompassing health, i.e., to “ensure healthy lives and promote well-being for all at all ages), and SDG5 (generally encompassing gender equality, i.e., to “achieve gender equality and empower all women and girls”). In essence, the broad scope of this study identifies the existence of a gender data gap in women’s healthcare [3], evidenced by contemporary literature, which, we claim, leads to gender bias and underrepresentation of women in the healthcare system. This means that women’s biological and socioeconomic differences and other considerations of intersectionality are not properly considered, resulting in inadequate health treatment or erroneous treatment protocols [4]. 

Advancements made in biomedical technologies include the proliferation of artificial intelligence (AI) applications in the healthcare dimension. The state-of-the-art shows that AI is increasingly being used for “diagnosis and treatment recommendations, patient engagement and adherence, and administrative activities” [5]. In medical research or life science companies and healthcare provision, some rudimentary AI systems have demonstrated their usefulness in performing tasks that can be easily automated; or where large amounts of patient data may require analysis or synthesis. The use of AI in the modern healthcare system and biomedicine has certainly enabled the transformation of techniques, tools, and methods of healthcare and medical delivery and treatment. For example, rehabilitative robotics are instrumental in helping patients re-acquire motor skills through the “implementation of efficient robot strategies” that allow these AI systems to behave as “trainers” for human sensory-motor systems [6]. Other uses of AI in biomedical health are also relevant for patient monitoring, genetic algorithms, optimization techniques [7], and other machine learning methodologies that would support the processing of large amounts of data for diagnostic purposes. Lau and Staccini state in their report of AI-related findings in 2018 that “recent AI developments showing promising results are data-driven approaches, specifically for clinician-facing applications such as image analysis and interpretation in radiology” [8]. These practical applications of AI in medicine, therefore, continue to serve an important role in a plethora of medical treatments, research, and diagnostics. 

In another stream of literature, we also find evidence of AI applications in the achievement of SDGs [9] and its potential positive implications on health, which may include women’s health [10]. While there is (separately), on the one hand, existing literature on some use of AI in women’s reproductive health (with a much more saturated concentration on AI in other aspects of health not relating to women), and on the other hand, literature that links health to the SDGs, there is no clear link between AI in women’s health holistically that links to the SDGs. The uniqueness of this approach also allows for an analysis of data vis-à-vis future studies: our future study will engage these findings with the lens of intersectionality [11] as a conceptual tool that situates privilege versus oppression, understanding these factors as barriers to achieving equality in women’s healthcare. Thus, we ask the following research question in this paper: Can applications of AI in women’s health expedite the achievement of gender equality in the health SDGs?

After critically analyzing the literature, we find that there appears to be a lack of representative data on women’s healthcare conditions more generally. However, through our SLR, we are confident that bridging this initial finding with contemporary AI applications in healthcare will allow us to make the hypothesis that ethically designed AI-driven technologies can contribute to women’s health. Particularly, the available data suggests that the proper enablement and ethical design of AI technologies for women’s healthcare can empower decision-making and informed health choices, which could expedite the achievement of SDG3 and SDG 5. This larger study is intended to form the scope of our future research in this field, leveraging on the findings that we now have to propose future foresight recommendations for women’s holistic healthcare. 

With this in mind, the next section of this paper elaborates on the methodology for this research, followed by the findings of the SLR. Finally, we conclude by discussing the key contributions of this study and mapping out the landscape of our next research study phase and future research scope in this area. 

## 2. Methodology 

Our SLR methodology in this paper is adopted to identify, select, critically appraise, and collate findings in prior relevant studies and identify the future research agenda related to our research topic. From the Scopus database, we collect papers from 2018 to 30 June 2022. To eliminate the possibility of selection bias [12], we used possible combinations of keywords related to “Health AI”, “Women Health”, “Gender Equality”, “Sustainable Development Goal 3”, and “Sustainable Development Goal 5”. The search is limited to published open-access, peer-reviewed journal articles written in English. The initial sample consists of 1864 documents. This sample size is consistent with other systematic literature reviews and bibliometric studies in the health and social sciences [13,14]. A filtering process has been carried out, consisting of independent reading of abstracts. A total of 343 articles were removed before screening because they were not relevant to the research question. A total of 1521 records were screened and 754 articles were removed because they did not include precise descriptions, were not full journal articles, did not match common search criteria followed in traditional SLR papers, and were published in languages other than English. Our final report, which was assessed for eligibility, left us with 767 documents. The search standard shows the appearance of the keywords in the “abstracts-title and keywords” and in the full text. In the next step, we read each article to conduct a qualitative analysis. From this, 165 papers were related to gender equality, 180 to AI in healthcare, 201 to sustainable development goals, 95 to different research approaches, and 126 to reviewing geographical locations. 

Figure 1 below reports the PRISMA of this study.

After adjusting the inconsistencies, we coded and tagged papers and included them in relevant clusters. For tagging, we used the relevant part of the text and mapped it with the keywords in the text of the papers. In Table 1 below, we report the clusters alongside the thematic markers and examples of the related keywords. 

## 3. Results

In this section, we critically analyze the clusters presented in Table 1 to provide some preliminary reflections on the proposed research question and map out an agenda for future research extending from this assessment. 

### 3.1. Cluster 1: Gender Equality

Gender equality can be measured in many ways, including political representation, employment characteristics, access to service, and standard indicators like the global gender gap index and the Gender Empowerment Measure [15]. However, there is limited attention devoted to the importance of gender equality and its contribution to enriching health studies [16]. To enrich the discussion on women’s health in medical science research, we adopt an interdisciplinary approach to examine gender equality in the attainment of the SDGs in health (SDG 3). 

The research is mainly focused on women’s health, but we aim to understand the cause behind the challenges faced by women of various ages in ages in various socioeconomic set-ups. In addition, to overcome these challenges in future research, we also discuss opportunities related to other gender classifications. Gender norms are often unspoken rules that determine attributes and behaviors. As it is often life-changing in nature, it can have significant effects on health that continue into adulthood and the next generation [17]. In many places, unequal gender norms mean that a girl may not have a choice in these life-altering decisions [18]. For girls, gender norms that emphasize girls’ sexual and reproductive capacity at the expense of education, capabilities, and agencies should be one target for prevention [17]. Table 2 summarizes the keywords that are considered important in existing studies. The keywords in Table 2 indicate that public policy drivers play an important role in empowering women and striving for equal opportunities in healthcare.

It is also important to study and understand women’s health within the sociocultural context in which they live, which sets expectations and boundaries on their experience [19]. Thus, we propose a need for further research to examine how the above reasons, which are responsible for women’s health, can be examined in certain socioeconomic contexts identified in the extant literature (hence, our proposed furtherance of this research using the lens of intersectionality). 

From the analysis, we find that a major health issue is related to pregnancy. Pregnancy risk factors, earlier age of sexual debut, exposure to sexual abuse, and a higher number of sexual partners are some key factors discussed in the literature [20,21]. A combination of health worker training, adolescent-friendly facility improvements, and broad information dissemination via the community, schools, and mass media are some important steps to develop reproductive health, especially in developing countries [22]. Applications of AI can play an instrumental role in creating the above opportunities to address the challenges associated with pregnancy. In addition, from our sample, we observe concerns around breast cancer treatment, prenatal care, mental health, and the impact of COVID-19. The findings of this study about the use of AI in addressing women’s health can assist and inform policy makers on how to achieve SDG 3 by 2030 alongside other SDGs. 

In the meantime, violence against women is recognized as a prominent public health problem; and sexual violence and victimization are associated with an increased risk of mental disorders [23]. This is consistent with what we find in our sample, i.e., sexual violence. Violence in intimate partner relationships is unfortunately common, which elevates risks for suicide and death by homicide. It should be given special consideration in the assessment and management of interpersonal violence [24]. Prevention and education are key to reducing sexual violence. Sexual violence victimization might reflect differences between the empowered and the vulnerable and can be related to any gender [25,26]. The Center for Disease Control and Prevention (2020) has proposed key components to reduce sexual violence; these include awareness and education, research, surveillance at all levels, improvement of public health systems, and proactive behaviors by individuals. Thus, additional research would be imperative to examine how AI could support initiatives to mitigate women’s health risks resulting from a diversity of socioeconomic backgrounds, impacting both physical and physiological health.

Our research also covers all women’s age groups (Table 3), associated health issues, and current measures by health workers and international organizations to improve the health conditions of women. Some 26.92% of the studied documents cover adolescent health issues, followed by newborns and infants (17.69%), and very elderly cohort age groups (16.15%). 

In the research papers, we discover that there is a variety of perspectives on adult women’s health. Malmusi et al. [27] suggest that women’s health inequalities emerge from an unequal distribution of social and family roles. In recent decades, the importance of biopsychosocial perspectives has been highlighted due to the influence of sociocultural variables on health [28], including patterns of employment, education, family, and social structure [27]. In addition, occupational and gender roles also play a significant role in women’s adulthood and health [29]. Esteban-Gonzalo et al. [30] in their research have identified that, in Spain, women with higher levels of education are less conditioned by gender roles and stereotypes, where educational environments seem to have some influence on their values and beliefs, keeping them in more tolerant and egalitarian positions. 

There is also a close association between traditional feminine roles and anxious and depressive symptomatology in adult women [30]. In developing regions like Africa, gender inequality in maternal health is indicated because of a lack of female autonomy, which poses a great risk to health. Wealthier households have low autonomy in household decision power and are more likely to be exposed to maternal health risks. Efforts have been made to lower women’s exposure to maternal mortality and morbidity, including interventions to alter prevailing gender norms that limit women’s active participation in decisions about their own health, especially during pregnancy and delivery [31]. Stenberg et al. [32], in their research, showed the three important integrated packages of essential health interventions which need to be scaled up: a reproductive health package, a maternal and newborn health package, and a child health package. Even though the Millennium Development Goal (MDG) era from 2000 to 2015 resulted in decent improvements in the health of mothers and children across the world, progress has been uneven. Children under the age of five are especially vulnerable to malaria, pneumonia, HIV, and tuberculosis [33]. Intracountry inequalities have increased, and more poor people are now living in high- and middle-income countries [34]. Almost nine million children are dying each year from preventable causes (including stillbirths), maternal mortality has lagged behind, and there is an unacceptably high number of children that have failed to achieve their growth potential due to low birth weight, stunting, and micronutrient deficiencies [35]. Newborn mortality can be prevented by providing bednet distribution, promotion of better hygiene, and oral rehydration solutions and antibiotics delivery by health workers [36]. 

In 2017, WHO launched a country-led campaign to improve the quality of care for mothers, newborns, and children called “Quality, Equity, Dignity”, and it now covers 11 countries with more than half the world’s births and perinatal deaths. Global and local responses are unsatisfactory, especially for children living in a conflict setting and/or belonging to vulnerable families [37]. Nurturing care from parents and other caregivers is critical for children (<5 years) who live in low- and middle-income countries [38]. Although the Lancet Commission has implemented a Global Strategy for Accelerated Action for the Health of Adolescents (AA-HA!), the latter remains the most neglected age group in health care [39,40]. Rising problems among adolescents are attributed to unhealthy diets, poor mental health, tobacco and alcohol use, early marriage and early pregnancy, and lack of hygiene. Notwithstanding, a UNICEF study [33] reports that self-harm is the leading cause of death for adolescents between the ages of 15 and 19. 

Community participation is important for successful health outcomes. WHO has recommended a participatory learning and action (PLA) approach in rural areas where newborn mortality is high [41]. In urban areas, community resource centers are working on improving maternal and child health [42]. Fottrell et al. [43], in their randomized trial research in Bangladesh, find that a 64% reduction in diabetic risk is possible through community mobilization. Recent research also shows the importance of community approaches in sanitary uptake in Mali and childhood development in Pakistan [44,45]. 

Based on these findings, we propose the undertaking of enhanced interdisciplinary research on women’s health, particularly using AI, to map out specific criteria that would encompass specific age groups and health problems that have a correlation with the socioeconomic trajectory of developed and developing countries. Targeting, for example, youth communities of girls, teenagers, and young adults as recipients of PLA approaches can be undertaken by AI systems to influence bespoke actions for SDG advancement.

### 3.2. Cluster 2: Health AI 

Health AI is a relatively recent emerging field. Documents derived from the Scopus database indicate a cluster of popularly employed AI techniques used in healthcare. AI in the health sector is not like AI in any other sector and deserves special consideration because firstly, people’s health is at stake; secondly, people are in a vulnerable position when in need of healthcare; thirdly, the collection of health data has dramatically increased in recent years, and finally; health data is historically littered with bias. Algorithms are trained based on big datasets with a huge amount of (personal) health data. High-quality health data is difficult to obtain [46], as it is often unstructured, inaccurate (because of measurement errors in self-reported health or medical records), and biased (because of lack of inclusive clinical data and underrepresentation of minorities) [47]. Underrepresented minorities could therefore risk a misdiagnosis. Biases in the training data can lead to discrimination and individual injury or even death. Racial bias, in particular, may lead to incorrect diagnoses and deepen existing socioeconomic inequality, something that is not considered in current regulations on medical technology. A lack of transparency and ability to explain algorithms or results and methodologies arising from algorithms threatens patients’ rights to information and informed consent to medical treatment, with vulnerable groups having potentially higher risks of harm as a result.

Linking to AI in health are digital health interventions that have achieved a new milestone in 2019 [48]. These now include a wide range of technologies across the spectrum of eHealth, mHealth, telemedicine, and even emerging areas of advanced computing such as “big data”, machine learning, and artificial intelligence. Accessibility to “big data” enables the “cognitive” computer to scan billions of bits of unstructured information, extract the information and recognize complex patterns. Computer-based, decision-support systems based on machine learning (ML) algorithms have the potential to develop medicine, decrease human resource costs, improve clinical workflow, and improve treatment choices. The radiomic features of AI technology capture the biological and pathophysiological information which provide rapid and accurate non-invasive biomarkers for cancer risk prediction, diagnostics, prognosis, treatment response monitoring, and tumor biology [49]. The application of machine learning in cardiovascular health care includes automated imaging interpretation, natural language processing and data extraction from electronic health records, and predictive analytics. These techniques are highly used in the interpretation of chest roentgenograms, electrocardiograms, echocardiograms, angiography, identification of patients with early heart failure, predicting mortality, or cardiovascular surgical procedures [50]. 

There are also both legal and ethical challenges in applying AI within the sphere of healthcare and medicine. Of key concern would be safety and transparency, algorithmic fairness and biases, data protection and privacy, cybersecurity, and informed consent to its use, amongst several other criteria [51]. The European Union’s (EU) effort to regulate the governance and trajectory of AI in Europe is represented by the EU AI Act, which is still currently undergoing approvals at the European Parliament level. There is some consternation as to the applicability of the EU AI Act, as systems are classified using risk-based approaches; this may not adequately capture all permutations of AI technologies [52]. Health as an issue is markedly absent from the EU AI Act, and hence, health AI may not adequately be covered under the Act. 

While there are tremendous efforts to apply health AI in a clinical setting, current risk prediction models based on AI technologies are still inadequate in many cases [53], one of which is dealing with labor pains during childbirth. A significant amount of research is still required to improve AI’s predictive ability in many aspects of women’s healthcare. Table 4, for example, demonstrates a small fraction of documents from our search, where percentage indicators are still extremely low when considering key issues such as global health and quality of life. This signals the necessity to ramp up efforts in making digital technologies in healthcare accessible to global communities, including women and underrepresented groups. We posit that an increase in digital literacy, knowledge, and resilience in the healthcare sector through AI technologies can also increase the advancement rates of the SDGs. 

### 3.3. Cluster 3: Sustainable Development Goals

As indicated in the introduction to this paper, our research understudies the possible acceleration of SDG3 and SDG5. The operative keywords in our documentation search in Scopus are chosen on the basis of the foundational values of the UN SDGs. In particular, our search underpins concepts of equality and public health as part of the human rights narrative. While the SGDs have been designed as a global effort, the documents in our SLR indicate that much more needs to be done to realize the goals. Manandhar et al. [54] have emphasized that interactions between SDGs 3 and 5 do indeed influence health outcomes; and more so, that specific domains of gender and health also interact with the broad frame of the UN SDGs 2030 agenda [54]. In our SLR, determinants such as social perspectives, healthcare systems, and education are likely the key contributors to health inequalities. 

SDG3 encompasses specific markers for gender-related health—these include “maternal mortality ratio, births attended by skilled health personnel, new human immunodeficiency virus (HIV) infections, satisfactory family planning, adolescent birth rates, and coverage of essential health services (which include reproductive and maternal health” [54]. The correlation with SDG5 is vital, as SDG5 seeks to increase collaborative measures and concrete actions to eliminate violence against women and girls. Both these gender and health SDGs are inextricably linked to human rights, and at the very least, these are necessary for one to exercise the right to health on a broad level. The right to health has been articulated in the 1946 WHO Constitution, the 1948 Universal Declaration of Human Rights, and other regional or supranational international human rights instruments [55].

The documents analyzed in Table 5 further enunciate the intertwining roles between legislators and public policy makers, as well as communities. For example, matters of public health, such as the recent COVID-19 situation, indicate that the pandemic exacerbated systemic inequalities in society, affecting minorities, the impoverished, and the underrepresented communities the most [56]. Moving forward in our research scope, it is clear that capacity building, national prioritization, institutional, and even socioeconomic and political agendas all play a part in advancing the UN SDGs. The findings from the literature also indicate that institutional and cultural differences in developing and developed countries also need to align “universal” meanings of human rights and equality in a bid to advance the SDGs in a meaningful way. 

### 3.4. Cluster 4: Research Approaches

From the Scopus database, we determine that there are three types of research approaches mainly in use in medical research. Cohort analysis and randomized controlled trials (RCTs) are the most used research approaches and cover 45.26% and 36.84% of the studied documents, respectively. Table 6 below indicate the findings on research approaches.

The term “cohort” in modern epidemiology refers to “a group of people with defined characteristics who are followed up to determine the incidence of, or mortality from, some specific disease, all causes of death, or some other outcome” [57]. A cohort study observes people in two or more groups, from exposure to outcome [58], and in this study design, the subject is followed up over a period of time. In the case of healthcare research, when there is evidence to suggest an association between exposure and an outcome, and the time interval between exposure and the development of outcome is reasonable, researchers allow cohort analysis. Cohort studies are the design of choice for determining the incidence and natural history of a condition. Due to their longitudinal design feature, one can look at disease progression and its natural history [59]. Cohort studies allow researchers to calculate the incidence rate, cumulative incidence, relative risk, and hazard ratio [59]. 

Following this, to understand the safety and efficacy of new treatments, RCTs are the best way to analyze data. RCTs are used to answer patient-related questions and are required by governmental regulatory bodies as the basis for approval decisions [60]. The basis of every RCT is the “study protocol that describes the medical/scientific background, the risk, benefit assessment, the study design, the study methods, and the overall planning, conduct and analysis” [61]. Some 17.89% of the documents and 31.56% of citation rates cover the usage of meta-analysis in medical research. This is a quantitative, formal, and epidemiological study design used to systematically assess previous research studies to derive conclusions about the specific issue [62]. Outcomes from a meta-analysis may include a more precise estimate of the effect of treatment or risk factors for disease [63]. The examination of heterogeneity in study results is also a critical outcome. The benefits of meta-analysis include a consolidated and quantitative review of a large and often complex, sometimes apparently conflicting body of literature. Meta-analysis, a statistical procedure that integrates the results of several independent studies, plays a central role in evidence-based medicine [64]. 

In contrast to medical science research, different research approaches in social science papers [65] may also be useful to incorporate interdisciplinary effectiveness in women’s healthcare studies. Based on these findings, we expect, in future research, to capture the impact of socioeconomic factors on women’s health through interdisciplinary dimensions. 

### 3.5. Cluster 5: Geographical Focus 

Following the literature review in our research based on geographical factors, we have categorized our results into ten location categories (indicated in Table 7 below).

Figure 2 shows the network of interconnected countries and Figure 3 explains the distribution of documents in the context of AI in health across the globe.

A large percentage of documents (23.31%) and citation rates (8.11%) cover AI implementations in healthcare in the United States (US); the US has a strong research association (77,538) with other countries as well. Within the US, the literature indicates that complications among individuals as a result of spine and bone disorders each year are increasing [66]. Researchers from the Perelman School of Medicine (Philadelphia) have constructed a neural network to output 2D segmentations for MR, CT, and X-ray imaging systems, all of which are helpful in treating spine and bone disorders [66]. Advances in machine learning (ML) technologies and their applications to intellectual and developmental disabilities (IDDs) and other neurological diseases have also increased over the last few decades.

There has been remarkable progress in biomarkers for clinical research and drug development [67] in this respect. The Food and Drug Administration (FDA) in the US released an action plan in 2021 (Office of the Commissioner, FDA, 2021) based on the use of AI as software [68] in medical devices and has mainly focused on device transparency, developing methods to evaluate and improve machine learning algorithms, and fostering patient-centered approaches. After the literature analysis, we find that health systems in the US are currently investing more in transformative AI applications to improve their competitive positioning, engage consumers, achieve profitable growth, and deliver personalized customer experiences. 

In the meantime, ethical and human rights issues of AI are the key drivers of the successful implementation of AI in healthcare in the UK and across Europe, especially in Germany and Spain. Researchers [69] discussed the ethical and human rights challenges adopted by the European Agency for AI. Researchers from the UK [70] have developed two machine learning models through knowledge engineering and training data from past cases. Among the region of South-East Asian countries, India (192 documents, 5243 citations, and 13,449 link strength) and China (339 documents, 4570 citations, and 13,449 link strength) are using advanced AI models and language to combat many diseases and these two countries have a strong link to each other as well (Figure 2). In the case of health applications development in India, researchers (Pandey et al., 2022) from the Department of Computational Biology in Delhi developed the “WashKaro” application to spread awareness amongst the population regarding Tuberculosis (TB) and COVID-19 information. This multi-pronged intervention uses conversational AI machine translation and natural language processing, which is available in all local languages. India is also accelerating the use of different sub-fields of AI to treat neonatal and infant death risk and breast cancer among women [71,72]. In China, the increasing population age over 65 in China has an impact on the public health sector, and these individuals need special care, treatment, equipment, and dependence on the younger generation. AI is shaping China’s healthcare and offering different diagnostic information, facilitating consultations with specialists, disease screening and prognosis prediction, optimizing medical resources, and carrying out medical tests in remote areas of rural China [73]. AI excellency is observed in the sector of robotics technology in China. It is now used in surgeries like orthopedic, laparoscopic, and neurologic procedures [73]. 

In the case of some African and Latin American countries, the literature review identified a significantly reduced number of documents based on AI and health. Due to financial constraints, governmental policies, and lack of resources, the general health infrastructures in these countries still need to cope with existing challenges. Thus, our findings indicate that there is a need for advanced research incorporating women’s health data from developing countries simultaneously with developed nations. The limitation of women’s health data from emerging countries raises questions about the successful implementation of the findings related to developed countries. 

## 4. Discussion 

From the sample, it is apparent that many AI applications in medicine and healthcare have largely focused on general cancer treatment [64] and cardiovascular disease [74]. In the meantime, the use of AI in women’s healthcare research is still limited, although starting to emerge, but is currently limited to mostly breast cancer or other gynecological research [75] As such, there is clearly an uneven representation of women’s healthcare conditions, relative to the large scope of other available general healthcare data. It should be noted that women’s healthcare encompasses much more than simply breast cancer or gynecological concerns [76]. In addition, we find while AI-driven technologies are currently deployed and are being used more widely in medical and healthcare research, and even in clinical settings, their use has remained limited to disease-specific interventions. This, of course, is likely due to capacity-building, the availability of funding and resources, and a need to broaden AI-related knowledge in healthcare. Hence, AI technological use in medicine and healthcare still lacks saturation. While it remains a developing techno-science field, studies have shown that this can, in fact, impact healthcare aspects that relate to women [77]. In this research, a key part of our theoretical contribution is to make use of this knowledge from the findings and to argue for regulators, policy makers, and relevant stakeholders to focus on the betterment and holistic encompassment of women’s healthcare; and use this intervention to develop a gender-equal society. In other words, women’s healthcare sustainable goals (beyond maternal and reproductive healthcare) should be discussed simultaneously with the gender equality goal. Because of the proliferation of AI-driven technologies in medicine and healthcare, we intend to show that the possibility of AI applications in women’s healthcare could assist nations in achieving and accelerating the gender equality and health SDGs defined by the United Nations. 

It is important to note that the existing literature is mainly context-based and focuses on one particular type of technology relevant to one specific issue associated with healthcare [75]. However, the applications of AI introduce new options for healthcare, which are yet to be explored in the literature. Nevertheless, this position is likely to change in the next decade, although it must be stated that new avenues of research relating to AI and healthcare must also move towards greater inclusion and representation.

## 5. Conclusions

Our theoretical contributions in empowering the use of AI technologies in women’s healthcare are consistent with new ways of thinking about gender equality in STEM [77]. Much of AI-relevant narratives in different regulatory environments highlight ethics in the design and implementation of such systems, including the EU AI Act (Proposal for a Regulation of the European Parliament and the Council Laying Down Harmonized Rules on Artificial Intelligence (Artificial Intelligence Act) and Amending Certain Union Legislative Acts, 2021), the OECD AI Principles [78], and even Singapore’s own national AI framework [79]. Even within most of these instruments, the inclusion of gender equality within the ethical framework is not given enough importance. 

To advance the achievement of sustainable goals, especially in health, we need active collaborative initiatives from the corporate sector, innovators, investment for innovation, and policy makers, amongst others. It is also imperative to note the key aims of a healthy and functioning health system. The findings of our research are of immense practical implications as we investigate how we might empower applications of ethically advanced technologies in women’s healthcare. This research is simply a minute indication that AI can be applied in women’s healthcare around the world (considering national resources, availability of skilled and trained personnel, and budget constraints), and so policy makers can incorporate advanced technologies such as AI in their strategic decision-making processes related to healthcare. Our findings also provide incentivization considerations to policy makers and regulators, as it shows a strong link between sustainable achievement in women’s health and gender equality.

This study is not free from limitations, like all other studies. The availability of granular data will allow us to develop clearer AI models, and if we map those machine findings with the stakeholders associated with women’s healthcare, we can finetune those findings accordingly. The current phenomenon of lack of data is one of the reasons to adopt the methodology based on SLR: to build a possible future framework to work on women’s healthcare and gender equality. In the future, we look towards conducting in-depth case studies on AI in women’s healthcare so that we may access more detailed and representative data. We should equally understudy other challenges and opportunities of using AI in healthcare and factor the equation of gender equality into the model.

## Figures and Tables

**Figure 1 healthcare-11-00401-f001:**
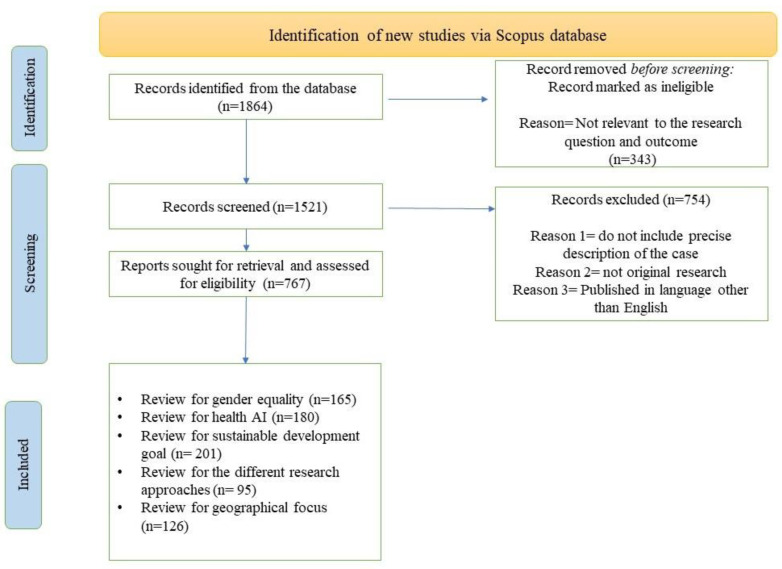
PRISMA Flow Diagram.

**Figure 2 healthcare-11-00401-f002:**
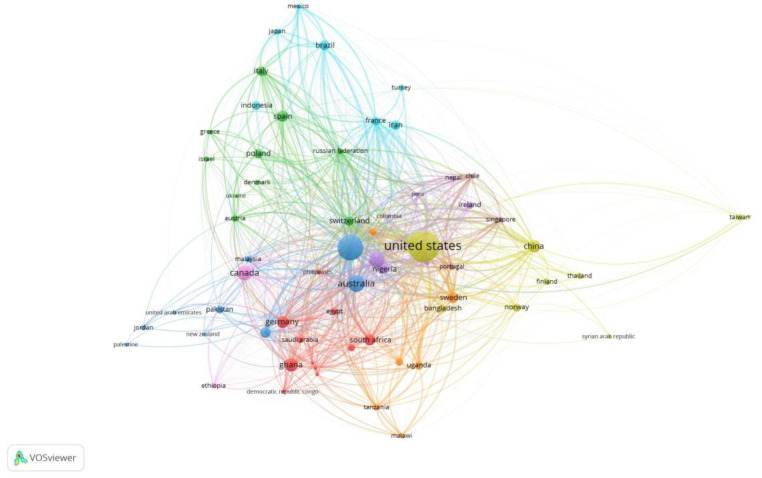
Geographic location cluster.

**Figure 3 healthcare-11-00401-f003:**
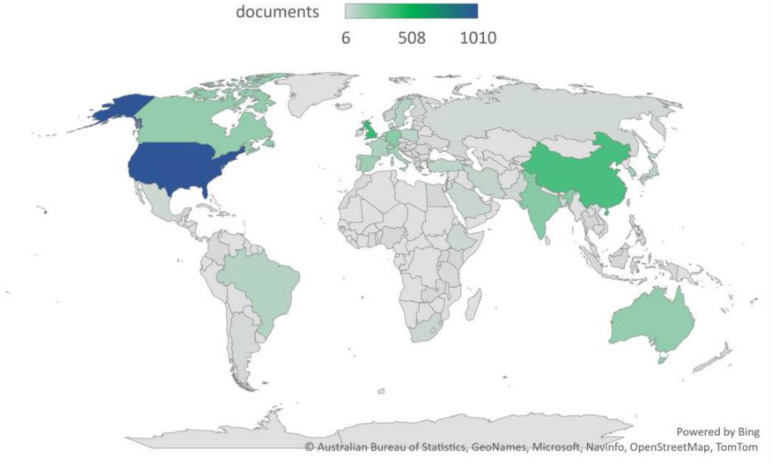
Distribution of documents based on AI implications on health across the globe. Source: Australian Bureau of Statistics, GeoNames, Microsoft, Navinfo, OpenStreetMap, TomTom.

**Table 1 healthcare-11-00401-t001:** System of Article Categorization: Gender Equality in Health AI.

Cluster	Thematic Marker	Keywords
1. Gender Equality	Women Other gender Age Socioeconomic factors	Female adults, health care personnel, empowerment, child health, maternal mortality, childhood mortality, sexual orientation, newborn, infant, child, adolescent, young adults, middle-aged, very elderly, mortality, prenatal care, healthcare cost, nursing, workforce, quality of life, health insurance and health promotion, social support, government, poverty, leadership, patient care, social policy
2. Health AI	Types of AI in health Gender Techniques of AI	Epidemiology, mental health, COVID-19, pregnancy, depression, prenatal care, cancer, artificial intelligence, female, prevention and control, retrospective study, machine learning, gender disparity
3. Sustainable Development Goals	SDG 3 SDG 5	Public health, health care delivery, healthcare services, healthcare policy, WHO, well-being, global health, quality of life, occupational health, equal participation; female adult, female, human rights
4. Research Approaches	Medical science Social science Other	Controlled study, major clinical study, cross-sectional study, qualitative research, health survey, outcome assessment, randomized controlled trial, quantitative analysis, thematic analysis, clinical trial, statistics and numerical data, cohort study, meta- analysis.
5. Geographical Focus	Developed countries Developing and emerging countries	United States of America, China, United Kingdom, India, Ghana, rural population, middle-income country, Nigeria, Africa

Source: Literature derived from the SCOPUS Database and prepared by Authors.

**Table 2 healthcare-11-00401-t002:** Focus on gender equality.

Keywords	Total Documents	Total Citation	% Document	% Citation
Equal participation	45	97	27.27	24.12
Empowerment	32	82	19.39	20.39
Violence and sexual exploitation	28	101	16.69	25.12
Public policy	60	122	36.36	14.92

Source: Authors’ calculation.

**Table 3 healthcare-11-00401-t003:** Focus on women’s age groups.

Keywords	Total Documents	Total Citation	% Document	% Citation
Newborns and infants	23	143	17.69	24.69
Child	11	89	8.46	15.37
Adolescent	35	159	26.92	27.46
Young adults	39	111	30	5.18
Middle age	9	56	6.92	9.67
Very elderly	13	21	16.15	3.62

**Table 4 healthcare-11-00401-t004:** Focus on Health AI.

Keywords	Total Documents	Total Citation	% Documents	% Citation
Global health	62	111	34.44	25.28
Quality of life	53	130	29.44	29.61
Artificial intelligence	35	97	19.45	22.09
Machine learning	30	101	16.66	6.83

**Table 5 healthcare-11-00401-t005:** Focus on Sustainable Development Goals.

Keywords	Total Documents	Total Citation	% Documents	% Citation
Public health	82	187	57.74	41.09
Equal participation	62	124	43.63	27.25
Human rights	59	144	29.35	31.64

**Table 6 healthcare-11-00401-t006:** Focus on Research Approaches.

Keywords	Total Documents	Total Citation	% Document	% Citation
Cohort study	43	111	45.26	27.07
Meta-analysis	17	131	17.89	31.56
Randomized controlled trial (RCT)	35	168	36.84	37.25

**Table 7 healthcare-11-00401-t007:** Focus on Geographical Location.

Keywords	% Document	% Citation	Total Link Strength
United States of America (USA)	23.31	28.11	61,101
Canada	8.21	13.33	42,759
United Kingdom (UK)	14.83	11.11	45,297
East Asia	9.01	7.22	42,115
South East Asia	27.15	25.79	21,479
Scandinavian, Central, and South European countries	4.83	3.22	10,433
Australia and New Zealand	3.78	3.77	17,837
Central Asia	6.44	4.13	25,357
Latin America	3.18	2.49	20,033
Africa	3.21	1.01	20,071

## Data Availability

This study is primarily a reanalysis of existing publicly available data as cited in the “References” section. Notwithstanding, in some sections of this publication, the data underpinning parts thereof can be accessed from Brunel University London’s data repository, Brunelfigshare here under a CCBY license: https://brunel.figshare.com/ publication (accessed on 21 July 2022), where it is supported by multiple datasets cited in the “References” section of this paper.

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
