# Peer review of "Accelerating UN Sustainable Development Goals with AI-Driven Technologies: A Systematic Literature Review of Women’s Healthcare"

_healthcare, 2023, doi:10.3390/healthcare11030401_

Round 1

Reviewer 1 Report

In this systematic review, Lau et al compile the currently available literature on the use of AI-based technologies in women’s healthcare and describe how this would help in achieving the sustainable development goals (SDGs) defined by the United Nations. They use a PRISMA flow chart to sequentially narrow down the relevant articles and then divided them into different thematic clusters. The authors claim that the findings from their systematic review can aid the government policies, accessibility, and regulating the environment in achieving the SDGs. However, these claims are not supported by strong results. Overall, there are several flaws in the way the data are presented and the manuscript is written that make this review article not suitable for publication in this journal in its current form.

   There are numerous grammatical and typing errors in the entire paper. Major English language editing is required.

   There is no need to define the word introduction as the first line in the introduction section.

    The authors should include a paragraph about what is known in terms of AI-based technologies in the introduction. They describe a lot about SDG but nothing about AI – which is an important part of their review.

    For systematic literature reviews, it is essential to define the exact key words used at the beginning of the search. The authors only give examples of the key words they used, which is not sufficient. How were the initial 3273 papers found?

   The entire paragraph from lines 98 to 110, seems like a rationale and justification for the study. That should not be a part of the methods. The authors can incorporate some points from this paragraph in the Introduction itself. Methods should only describe what was exactly done in the paper.

    Figure 1 showing the PRISMA flow diagram needs to be cited in the first place where the process of selection is described – in the paragraph starting at line 111.

    Figure 1 describes 5 categories of papers that were included in this article, along with exact numbers. However, they describe 8 clusters in the results. Why a discrepancy between 5 and 8 clusters/categories?

  The authors should improve the order in which they have presented the clusters. If this SLR is about the use of AI technology in women’s health, they have to give priority to clusters like AI techniques in health research, women’s age groups and types of medical treatment, and so on. Why the “health and well-being” cluster was chosen as the first cluster is unclear.

   The entire results section lacks structure and does not flow properly. The authors identify 3-5 sub-points under each cluster and give the percentage of papers that discuss each of those- this is presented well in a table format. But the text does not go through each of those sub-points in an organized and clear way.

1   In some clusters, like types of medical treatment, the authors are describing how AI is being used for prostate cancer. How is that relevant to women’s health? This also raises the question whether the papers included in this SLR cover both male and female patients or do they only cover male patients in some scenarios. Because this SLR is on the use of AI technology in women’s health, it is important that the authors include articles that have female patients.

    What is the point of having Table 10 that describes the analysis of the findings? For most of the clusters, the discussion/conclusion just states that there is still a gap in knowledge or that detailed information does not exist. These simple points can be summed up in a few sentences itself. The table doesn’t add any value to the manuscript.

Lastly, there is a lot of redundancy in the entire manuscript. It should be significantly shortened and the text should be made concise and clear.

Author Response

24th January 2023

To the Editor in Chief and Reviewer 1,

Thank you for giving us an opportunity to revise the paper.  We thank the Reviewers for their suggestions and comments which, we believe, have led to an improved and clearer version of the manuscript.

We have tried to take all their comments into account. We have also taken the opportunity to improve some other parts of the manuscript, and corrected typos, grammar, misprints, and the like. The manuscript has been professionally proofread by a native English speaker.

In response to the technical comments we have received, please find below our responses addressing each point raised by the Reviewers.

Yours sincerely,

Pin Lean Lau (corresponding author)

(*Reviewer comments/feedback are indicated in blue font; our replies are indicated in automatic black font).

Reply to Reviewer 1

  1. In this systematic review, Lau et al compile the currently available literature on the use of AI-based technologies in women’s healthcare and describe how this would help in achieving the sustainable development goals (SDGs) defined by the United Nations. They use a PRISMA flow chart to sequentially narrow down the relevant articles and then divided them into different thematic clusters. The authors claim that the findings from their systematic review can aid the government policies, accessibility, and regulating the environment in achieving the SDGs. However, these claims are not supported by strong results. Overall, there are several flaws in the way the data are presented and the manuscript is written that make this review article not suitable for publication in this journal in its current form.

We thank you for your comments, which has provided us with greater clarity on how we present our findings. In our revisions, we also undertook additional reviews to support our hypotheses, and removed clusters that we no longer deemed necessary to the narrative. An extensive revision of the manuscript was carried out. Responses to each point raised can be found below.

  1. There are numerous grammatical and typing errors in the entire paper. Major English language editing is required.

We regret that these had been significantly overlooked in the editing process. Major revisions have been undertaken, and the manuscript has been additionally proofread by a native English speaker.

  1. There is no need to define the word introduction as the first line in the introduction section.

In the revised manuscript, similar definitions such as this, have been removed. The introduction has also been revamped, in the hopes that this provides precision and clarity.

  1. The authors should include a paragraph about what is known in terms of AI-based technologies in the introduction. They describe a lot about SDG but nothing about AI – which is an important part of their review.

       This has now been undertaken. In the introduction, as per this comment, a paragraph has been included on AI-driven technologies. The paragraph explains the state-of-the-art of AI technologies in medicine and healthcare, and ties its proliferation to the literature review that identifies its deployment in healthcare settings. Please see lines 51-70.

  1. For systematic literature reviews, it is essential to define the exact key words used at the beginning of the search. The authors only give examples of the key words they used, which is not sufficient. How were the initial 3273 papers found?

We have revisited our systematic literature review, removing sections/clusters that we believe do not now add value to the discussion (also considering the comments that have been made in the following points). For example, we have provided thematic markers of the keywords in a new table, followed by the definitional scope of what those keywords would encompass. Following our revised review, we also significantly reduced our sample size, and believe this is aligned with sample sizes that are ‘usual’ for a systematic literature review. Please see revised methodology (lines 99-118) and Table 1 (line 126).

  1. The entire paragraph from lines 98 to 110, seems like a rationale and justification for the study. That should not be a part of the methods. The authors can incorporate some points from this paragraph in the Introduction itself. Methods should only describe what was exactly done in the paper.

We thank you for this suggestion – we have followed the advice. Some of these points have now been extracted and included in the introduction. The methodology section has also been refined and rewritten (please see lines 99-118).

  1. Figure 1 showing the PRISMA flow diagram needs to be cited in the first place where the process of selection is described – in the paragraph starting at line 111.

This has now been done.

  1. Figure 1 describes 5 categories of papers that were included in this article, along with exact numbers. However, they describe 8 clusters in the results. Why a discrepancy between 5 and 8 clusters/categories?

This has now been corrected, following revisions in the manuscript. We have now limited the clusters to 5 main clusters, with specific thematic clusters and a scope/range of the relevant keywords. All previous clusters have been removed. Please see the new clusters in Table 1 (line 126).

  1. The authors should improve the order in which they have presented the clusters. If this SLR is about the use of AI technology in women’s health, they have to give priority to clusters like AI techniques in health research, women’s age groups and types of medical treatment, and so on. Why the “health and well-being” cluster was chosen as the first cluster is unclear.

This has now been changed, following revisions in the manuscript. To clarify, the clusters were initially presented in no particular order of importance – but we understand how it might be viewed that way, and we thank you for your observation and suggestion.

  1. The entire results section lacks structure and does not flow properly. The authors identify 3-5 sub-points under each cluster and give the percentage of papers that discuss each of those- this is presented well in a table format. But the text does not go through each of those sub-points in an organized and clear way.

Following the extensive revisions, we hope that the results section now read more smoothly. In addition to referring to the literature for each cluster, we also provide a slightly more in-depth analysis and our opinions as to the implications of these readings. We hope this provides the clarity and organization that is needed.

  1. In some clusters, like types of medical treatment, the authors are describing how AI is being used for prostate cancer. How is that relevant to women’s health? This also raises the question whether the papers included in this SLR cover both male and female patients or do they only cover male patients in some scenarios. Because this SLR is on the use of AI technology in women’s health, it is important that the authors include articles that have female patients.

Due to the revisions we have undertaken in the manuscript, these clusters are no longer part of the manuscript. For clarity, we wanted to demonstrate that much of AI in healthcare did not adequately focus on women’s healthcare, save for reproductive issues – and the types of medical treatment indicated AI’s most common usage in these areas. To additionally clarify, whilst there are studies that include female patients, this is presently the issue that plagues the quality of most medical data encompassing general disease diagnosis and treatment because of the under-representation of women in these studies (including but not limited to clinical trials). In any case, these clusters are no longer relevant in our revised manuscript, and we had sought to focus on very specific aspects of women’s healthcare considerations instead.

  1. What is the point of having Table 10 that describes the analysis of the findings? For most of the clusters, the discussion/conclusion just states that there is still a gap in knowledge or that detailed information does not exist. These simple points can be summed up in a few sentences itself. The table doesn’t add any value to the manuscript.

We thank you for your feedback. Indeed, Table 10 has been completely removed now, following the extensive revisions to the manuscript. The simple points have been summed up in the results section for each cluster, with a more detailed analysis of the literature, and the wider policy debates surrounding health equalities.

  1. Lastly, there is a lot of redundancy in the entire manuscript. It should be significantly shortened and the text should be made concise and clear.

With the current revisions made to the manuscript, we believe that this will address your concern. Most parts have been significantly reduced, redundant and pedantic parts have been removed, and we have tried to convey the points more clearly and directly. The total number of words has been significantly cut down from the initial 13,000++ words to now 9,000++ words.

Thank you very much.

Reviewer 2 Report

This reviewer finds the paper well written and interesting for the AI Community. However a relevant issue to be addresed is that the purpose of the paper is not clearly stated. In line 85 authors claim: "The purpose of the study is to analyse contributions of artificial intelligence (AI) in the advancement of health sustainable development goals." Nonehtless, in lines 127-129 authors state: "The focus of the study is to determine if AI has potential to address social challenges 127 unique to global health [10], especially related to women and accelerate the achievement 128 of sustainable development goals [11] with special focus on SDG 3 and SDG 5." Pease clarify this.

Moreover, conclussions must be focused on the authors' findings regardind AI Driven Technologies for Women’s Healthcare, as title of the paper addresses.

Author Response

To the Editor in Chief and Reviewer 2,

Thank you for giving us an opportunity to revise the paper.  We thank the Reviewers for their suggestions and comments which, we believe, have led to an improved and clearer version of the manuscript.

We have tried to take all their comments into account. We have also taken the opportunity to improve some other parts of the manuscript, and corrected typos, grammar, misprints, and the like. The manuscript has been professionally proofread by a native English speaker.

In response to the technical comments we have received, please find below our responses addressing each point raised by the Reviewers.

Yours sincerely,

Pin Lean Lau (corresponding author)

(*Reviewer comments/feedback are indicated in blue font; our replies are indicated in automatic black font).

Reply to Reviewer 2

  1. This reviewer finds the paper well written and interesting for the AI Community. However, a relevant issue to be addressed is that the purpose of the paper is not clearly stated. In line 85 authors claim: "The purpose of the study is to analyse contributions of artificial intelligence (AI) in the advancement of health sustainable development goals." Nonetheless, in lines 127-129 authors state: "The focus of the study is to determine if AI has potential to address social challenges 127 unique to global health [10], especially related to women and accelerate the achievement 128 of sustainable development goals [11] with special focus on SDG 3 and SDG 5." Please clarify this.

We have revised the manuscript extensively, with these points having been removed – and we believe that it now reads with greater clarity and precision. We thank the reviewer for these comments, and note the initial discrepancy in the purpose and focus of the study. As such, through our comprehensive revision process, we have now aligned the focus of the paper accordingly, with the literature review. You will note that the focus has now been pivoted to determine if AI applications in healthcare adequately represent women’s healthcare more specifically; and if not, we make the hypotheses that its future deployment can bring benefits. We trust that this is now addressed in our revised manuscript.

  1. Moreover, conclusions must be focused on the authors' findings regarding AI Driven Technologies for Women’s Healthcare, as title of the paper addresses.

We thank you for the comment, and agree that greater clarity was needed to emphasize the findings. In this regard, our revised manuscript, also analyzes each of the thematic clusters (based on exact keywords) – and how these clusters, etc, factor into the equation regarding AI-driven technologies. In each mini conclusion, so to speak, (ie in each cluster), we substantially revised the manuscript to also incorporate our analyses. We believe this lends a more well -rounded dimension to the findings/results, as well as ties in to the over-arching idea that AI-driven technologies, whilst currently under-represented in women’s health, must be further undertaken to help accelerate the SGDs and bring benefits in the general realm of women’s healthcare.

Thank you.